# Oracle Inequalities for Model Selection
# in Offline Reinforcement Learning

**Jonathan N. Lee**
Stanford University
jnl@stanford.edu

**George Tucker**
Google Research
gjt@google.com

**Ofir Nachum**
Google Research
ofirnachum@google.com

**Bo Dai**
Google Research
bodai@google.com

**Emma Brunskill**
Stanford University
ebrun@cs.stanford.edu

## Abstract

In offline reinforcement learning (RL), a learner leverages prior logged data to learn a good policy without interacting with the environment. A major challenge in applying such methods in practice is the lack of both theoretically principled and practical tools for model selection and evaluation. To address this, we study the problem of model selection in offline RL with value function approximation. The learner is given a nested sequence of model classes to minimize squared Bellman error and must select among these to achieve a balance between approximation and estimation error of the classes. We propose the first model selection algorithm for offline RL that achieves minimax rate-optimal oracle inequalities up to logarithmic factors. The algorithm, MODBE, takes as input a collection of candidate model classes and a generic base offline RL algorithm. By successively eliminating model classes using a novel one-sided generalization test, MODBE returns a policy with regret scaling with the complexity of the *minimally complete* model class. In addition to its theoretical guarantees, it is conceptually simple and computationally efficient, amounting to solving a series of square loss regression problems and then comparing relative square loss between classes. We conclude with several numerical simulations showing it is capable of reliably selecting a good model class.[1]

## 1 Introduction

Model selection is a fundamental task in supervised learning and statistical learning theory. Given a sequence of model classes, the goal is to optimally balance the approximation error (bias) and estimation error (variance) offered by the potential model class choices, even though the best model class is not known in advance. Model selection algorithms are extremely well-studied in learning theory (Massart, 2007; Lugosi and Nobel, 1999; Bartlett et al., 2002; Bartlett, 2008), and methods like cross-validation have become essential steps for practitioners.

In recent years, interest has turned to model selection in decision-making problems like bandits and reinforcement learning. A number of theoretical works have studied the *online* setting (Agarwal et al., 2017; Foster et al., 2019; Pacchiano et al., 2020; Lee et al., 2021a; Modi et al., 2020; Chatterji et al., 2020; Muthukumar and Krishnamurthy, 2021). Similar to the bias-variance balance in supervised learning, these algorithms typically aim to select the model class with smallest statistical complexity that contains the true model. Despite these recent efforts, the current understanding of model selection in *offline* (or batch) reinforcement learning (RL) is comparatively nascent. Offline RL is a paradigm where the

---

[1] Supplementary material is available at: https://sites.google.com/stanford.edu/offline-model-selection.

36th Conference on Neural Information Processing Systems (NeurIPS 2022).

learner leverages prior datasets of logged interactions with the environment (Lange et al., 2012; Levine et al., 2020). The learner is tasked with returning a good policy without further environment interaction. As has been acknowledged in several recent papers (Xie and Jiang, 2021; Mandlekar et al., 2021; Kumar et al., 2021), one of the major challenges preventing widespread deployment of offline RL algorithms in the real world is the lack of algorithmic tools for model selection, evaluation, and hyperparameter tuning. In experimental settings, researchers typically evaluate candidate learned models by using online rollouts of the policies after learning with offline data. However, such approaches are not feasible in many real world settings where the entire process of producing a single policy must be conducted only on the offline dataset, due to complications such as logistics, safety, or performance requirements.

In recent years, this problem has been recognized as a major deficiency in the field and a number of efforts have been made to remedy it. On the empirical side, several researchers have proposed workflows and general heuristics specifically addressing this problem (Kumar et al., 2021; Tang and Wiens, 2021; Paine et al., 2020). However, all have noted that solutions designed to evaluate or select models typically have their own hyperparameters and modeling choices. Consider, for example, applying off-the-shelf offline policy evaluation (OPE) methods (Precup, 2000; Thomas and Brunskill, 2016). These typically require some function approximation of their own. Thus, rather than solving the problem, naively using OPE just shifts the burden of model selection to the OPE estimator. Similarly, recent efforts to solve model selection in *online* bandits and RL are inapplicable as they almost universally require interaction with the environment (Foster et al., 2019; Pacchiano et al., 2020; Lee et al., 2021a). The solution to the *offline* problem seems to require new ideas.

On the theoretical side, there is also significant motivation for devising model selection algorithms as there is growing evidence suggesting that strong conditions on the function class[2] are necessary to achieve non-trivial guarantees in offline RL in the worst case (Foster et al., 2021; Zanette, 2021; Wang et al., 2020). Perhaps the most widely used and recognized condition is completeness (Munos and Szepesvári, 2008; Antos et al., 2008; Chen and Jiang, 2019) which essentially says that $\mathcal{T}f \in \mathcal{F}$ for any $f \in \mathcal{F}$, where $\mathcal{T}$ is the Bellman operator and $\mathcal{F}$ is the model class.[3] Unsurprisingly, completeness plays an important role in the proofs of many value-based offline RL algorithms since sample efficient results are provably impossible without it (in the absence of additional assumptions – see Xie and Jiang (2021); Zhang and Jiang (2021)). Despite the growing realization of the importance of these conditions, there seems to be comparatively little work addressing the problems of identifying complete model classes or certifying sufficient conditions for sample efficient offline RL. Lee et al. (2021b) considered the problem of model selection in the offline setting with the intent of addressing some of the aforementioned issues. It was shown that full model selection (competitive with an oracle that has knowledge of the best model class) is impossible in general in offline reinforcement learning. They proposed several relaxations to achieve weaker oracle inequalities, but these were limited to contextual bandits with linear model classes where there is no issue of completeness. The question of whether any similar results are possible for full offline reinforcement learning with general function classes has remained open.

## 1.1 Contributions

**Theoretical Guarantees** In this paper, we give the first rate-optimal model selection algorithm for offline RL with value function approximation. We begin by summarizing known results for a single model class using value-based methods. For any individual model class $\mathcal{F}$ that satisfies completeness and an offline dataset of $n$ samples with sufficient coverage, the gold-standard regret bound is $\tilde{\mathcal{O}}\left(\sqrt{\text{COMP}(\mathcal{F})/n}\right)$[4] where $\text{COMP}(\mathcal{F})$ denotes the statistical complexity of $\mathcal{F}$. This is achieved, for example, by Fitted Q-Iteration (FQI) (Chen and Jiang, 2019). Clearly, one would like $\text{COMP}(\mathcal{F})$ be as small as possible to achieve a tighter bound.

We consider the model selection problem where we are given an offline dataset of $n$ samples and a nested sequence of $M$ model classes $\mathcal{F}_1 \subseteq ... \subseteq \mathcal{F}_M$. We investigate the following question: *Can*

---

[2]That is, conditions sufficient for supervised learning, like realizability, tend not to be sufficient on their own for offline RL.

[3]$\mathcal{F}$ is a model class meant to estimate $Q$-functions. It consists of functions mapping state-action pairs to value predictions. The Bellman operator applied to $f \in \mathcal{F}$ pointwise is defined as $\mathcal{T}f(x,a) = r(x,a) + \max_{a'}\mathbb{E}_{x'|x,a}f(x',a')$.

[4]For clarity, $\tilde{\mathcal{O}}$ omits dependence on certain parameters such as the horizon $H$, distribution mismatch factors, number of classes $M$, failure probability $\delta$, log factors, and constants.

*we achieve a model selection guarantee for offline RL with regret scaling with the complexity of the smallest complete model class?*

We present a novel and conceptually simple algorithm, MODBE, that achieves regret scaling with the complexity of the smallest class satisfying completeness without knowledge of this class *a priori*.

**Theorem 1.** *(informal version of Corollary 1) Given an offline dataset of $n$ samples and nested model classes $\mathcal{F}_1 \subseteq ... \subseteq \mathcal{F}_M$, MODBE outputs $\hat{\pi}$ such that $\textbf{Reg}(\hat{\pi}) = \tilde{\mathcal{O}}\left(\sqrt{\text{COMP}(\mathcal{F}_{k_*})/n}\right)$ where $k_* = \min\{k \in [M] : \mathcal{F}_k \text{ is complete}\}$.*

A guarantee of this nature is typically known as an *oracle inequality* since an oracle with knowledge of the "best" model class ahead of time could simply choose it. We remark that this oracle inequality is rate-optimal in $\text{COMP}(\mathcal{F}_{k_*})$ and $n$, showing that we do not have to sacrifice efficiency for adaptivity. This is in contrast to some other works in model selection for decision-making where this unfortunate efficiency-adaptivity trade-off has been observed (Foster et al., 2019; Pacchiano et al., 2020; Xie and Jiang, 2021). In Appendix A, we discuss how the nestedness condition is necessary.

We also provide a robustness result for model selection (Theorem 3): if no models are Bellman complete (that is, $k_*$ does not exist), MODBE obtains $\textbf{Reg}(\hat{\pi}) \leq \tilde{\mathcal{O}}\left(\min_{k \in [M]} \sqrt{\xi_k + \text{COMP}(\mathcal{F}_k)/n}\right)$ where $\xi_k$ is a measure of the *global* completeness error of $\mathcal{F}_k$.[5] Our results show that, while some model selection problems remain elusive without further assumptions, strong rate-optimal oracle inequalities are still possible under standard offline RL assumptions even without knowledge of the best classes in advance.

**Technical Highlights.** The key to achieving the near optimal regret rate is to achieve the near optimal excess risk rate of the squared Bellman error (which is of order $\tilde{\mathcal{O}}(\text{COMP}(\mathcal{F}_{k_*})/n)$). To do this, MODBE iteratively compares the relative effectiveness of two candidate model classes by employing a hypothesis test that compares the difference of their estimated risks to a *one-sided* generalization bound. The fact that the test leverages only the one-sided generalization bound is crucial: using easier two-sided bounds (e.g. from uniform deviation bounds on risk estimators) leads to a squared Bellman error rate of $\tilde{\mathcal{O}}\left(\sqrt{\text{COMP}(\mathcal{F}_{k_*})/n}\right)$, which translates to a slow $\tilde{\mathcal{O}}((\text{COMP}(\mathcal{F}_{k_*})/n)^{1/4})$ regret rate. Instead the one-sided generalization error allows us to ultimately obtain the optimal $\tilde{\mathcal{O}}\left(\sqrt{\text{COMP}(\mathcal{F}_{k_*})/n}\right)$ regret rate.

**Practical Results.** In practice, MODBE can be instantiated with *any* base offline RL algorithm that attempts to minimize squared Bellman error, including but not limited to FQI. MODBE is also computationally efficient, requiring $\mathcal{O}(Hk_*M)$ calls to an empirical squared loss minimization oracle and $\mathcal{O}(k_*)$ calls to the base offline RL algorithm. In Section 5, we demonstrate the effectiveness of MODBE on several simulated experimental domains. We use neural network-based offline RL algorithms as baselines and show that MODBE is able to reliably select a good model class.

## 1.2 Additional Closely Related Work

Several prior works have specifically set out to address the model selection problem from a theoretical perspective, as we do here. Lee et al. (2021b) formalized the end-to-end model selection problem for offline RL where, given nested model classes, the goal is to produce a regret bound competitive with an oracle that has knowledge of the optimal model class. Their positive results, however, were limited only to linear model classes for contextual bandits; ours apply to sequential settings. An earlier work by Farahmand and Szepesvári (2011) had partially addressed our problem but made several restrictive assumptions such as a known generalization bound that underestimates the approximation error (which is generally unknown); our algorithm only relies on commonly known quantities. Another notable work is the BVFT algorithm of Xie and Jiang (2021). While initially designed for general policy optimization, BVFT can be applied to model selection (Zhang and Jiang, 2021) but it incurs a slow $1/n^{1/4}$ regret rate in theory (compared to our $1/n^{1/2}$) and requires a stronger data coverage assumption. One advantage of BVFT is that it can be used more generally to tune hyperparameters beyond the selection of model classes. However, the specialization of our algorithm to model selection enables the stronger guarantees. Thus, we view the two algorithms as complementary. Jiang et al. (2015) studied abstraction selection between nested state abstractions of increasing granularity; however,

---

[5]See Section 3.1 for a precise definition.

this eschews problems specific to value function approximation setting. Hallak et al. (2013) studied a similar abstraction problem, giving only asymptotic guarantees. In Section 3.2, we will discuss in more detail why several seemingly natural approaches to model selection do not produce satisfactory results.

## 2 Preliminaries

**Notation**  For any $n \in \mathbb{N}$, we let $[n] = \{1,...,n\}$. The notation $a \lesssim b$ implies that $a \leq Cb$ for some absolute constant $C > 0$. We will use $C, C_1, C_2 ... > 0$ to denote absolute constants (independent of problem parameters). For a set $A$, $\Delta(A)$ denotes the set of distributions over $A$.

We consider the finite-horizon Markov decision process $\mathcal{M}(\mathcal{X}, \mathcal{A}, H, \mathbb{P}, r, \rho)$ where $\mathcal{X}$ is the (potentially infinite) state-space, $\mathcal{A}$ is the action space, $H$ is the length of the horizon, $\mathbb{P} : \mathcal{X} \times \mathcal{A} \to \Delta(\mathcal{X})$ is the transition kernel, $r : \mathcal{X} \times \mathcal{A} \to [0,1]$ is a deterministic reward function, and $\rho \in \Delta(\mathcal{X})$ is an initial state distribution. A learner interacts with the MDP by proposing an $H$-step policy $\pi = (\pi_h)_{h \in [H]}$ where each $\pi_h : x \mapsto \pi_h(\cdot|x)$ maps $x \in \mathcal{X}$ to a distribution over actions in $\Delta(\mathcal{A})$.[6] At step $h = 1$, $x_1$ is drawn according to $\rho$. Then at step $h \in [H]$, the agent observes $x_h$, draws $a_h$ according to $\pi_h(\cdot|x_h)$ observes reward $r(x_h, a_h)$ and the MDP transitions to $x_{h+1}$ according to $\mathbb{P}(\cdot|x_h, a_h)$. For a policy $\pi$, we let $P_h^\pi(x,a)$ and $P_h^\pi(x)$ denote the marginal state-action and state densities of $\pi$ respectively at step $h$.

Following standard definitions, we let $V_h^\pi : \mathcal{X} \to \mathbb{R}$ denote the value function of $\pi$ at step $h \in [H]$ which is given by $V_h^\pi(x) = \mathbb{E}_\pi \left[ \sum_{s \geq h} r(x_s, a_s) \Big| x_s = x \right]$. Here, the expectation $\mathbb{E}_\pi$ is over trajectories under $\pi$ with $a_h \sim \pi_h(\cdot|x_h)$. Similarly, the action-value function $Q_h^\pi : \mathcal{X} \times \mathcal{A} \to \mathbb{R}$ is defined as $Q_h^\pi(x,a) = \mathbb{E}_\pi \left[ \sum_{s \geq h} r(x_s, a_s) \Big| x_s = x, a_s = a \right]$. The optimal policy (which exists under mild conditions when $H$ is finite (Sutton and Barto, 2018)) is denoted by $\pi^*$ and this maximizes $V_h^\pi(x)$ for all $x$ and $h$. The average value of a policy $\pi$ is given by $v(\pi) := \mathbb{E}_{x \sim \rho}[V_1^\pi(x)]$. Finally, we define the Bellman operators: $T_h^\pi Q(x,a) = r(x,a) + \mathbb{E}_{x' \sim P(\cdot|x,a), a' \sim \pi_{h+1}(\cdot|x')}[Q(x',a')]$ and $T_h^* Q(x,a) = r(x,a) + \mathbb{E}_{x' \sim P(\cdot|x,a)}[\max_{a' \in \mathcal{A}} Q(x',a')]$. Note that the values of $v(\pi)$, $V_h^\pi$, and $Q_h^\pi$ are always in $[0,H]$ due to the constraint on $r$. For convenience, we denote the $Q$ function of the optimal policy as $Q^* = Q^{\pi^*}$.

We consider the setting where the learner is provided with a model class $\mathcal{F} \subseteq (\mathcal{X} \times \mathcal{A} \to [0,H])$ to estimate action value functions at each step. For exposition, we assume this model class is *finite*; however, it is straightforward to extend to infinite settings with appropriate complexity measures. For simplicity, we will assume that the learner uses the same $\mathcal{F}$ for each timestep $h \in [H]$ but this is trivially extended. We assume that $0 \in \mathcal{F}$ and we always write $f_{H+1} = 0$. For any function $f \in \mathcal{X} \times \mathcal{A} \to [0,H]$, we define the argmax policy $\pi_f(x) = \operatorname{argmax}_{a \in \mathcal{A}} f(x,a)$. We will also write $f(x) = \max_{a \in \mathcal{A}} f(x,a)$.

### 2.1 Offline Reinforcement Learning

The distinguishing feature of the offline (or batch) RL is that we assume that the learner is provided with a dataset $D$ of example transitions in the MDP. The learner itself is not permitted to interact in the environment. The objective is to produce a good policy $\hat{\pi}$ using only data from the dataset $D$.

Formally, the dataset decomposes as $D = (D_h)_{h \in [H]}$ for each timestep where $D_h = \{(x,a,r,x')\}$ consists of tuples of transitions and incurred rewards. We assume $D_h$ contains $n$ datapoints that are sampled i.i.d from a fixed marginal distribution $\mu_h \in \Delta(\mathcal{X} \times \mathcal{A})$ and the data are independent across timesteps $h$. That is, there are $Hn$ datapoints total. For example, the data could be generated from $h$-step state-action distribution of a behavior policy $\pi^b$ so that $\mu_h(x,a) = P_h^{\pi^b}(x,a) = \pi_h^b(a|x) P_h^{\pi_b}(x)$.

For $f,g \in (\mathcal{X} \times \mathcal{A} \to \mathbb{R})$, we use the notation $\|f - g\|_{\mu_h}^2 = \mathbb{E}_{\mu_h}\left[ (f(x,a) - g(x,a))^2 \right]$. The average squared Bellman error under $\mu$ at state $h$ with respect to $f,g$ is $\|f - T_h^* g\|_{\mu_h}^2$. Following classical conventions (Munos and Szepesvári, 2008; Duan et al., 2021), we make a concentrability assumption that the data distribution $\mu$ has good coverage over the MDP for all reachable state-actions.

**Assumption 1.** *There exists a constant $\mathcal{C}(\mu) > 0$ such that* $\sup_{h,x,a,\pi} P_h^\pi(x,a)/\mu_h(x,a) \leq \mathcal{C}(\mu)$.

Concentrability is a structural assumption and it is widely regarded as perhaps the most standard assumption when studying offline RL problems (Foster et al., 2021). We remark that recent theoretical

---

[6]With some abuse of notation, for deterministic $\pi_h$ we write $a = \pi_h(x)$ to denote its highest-probability action.

works have striven to weaken this condition via pessimistic methods (Liu et al., 2020; Jin et al., 2021; Xie et al., 2021; Uehara and Sun, 2021). However, Theorem 2 of Lee et al. (2021b) shows that model selection bounds of this type are not possible even in contextual bandits and even though the single model class bounds are possible. As a result, we will not consider this refinement in the present paper.

In this offline setting, the learner aims to use $D$ and $\mathcal{F}$ to produce a policy $\hat{\pi}$ so as to minimize the regret, which measures the difference in average value between the optimal policy $\pi^*$ and $\hat{\pi}$:

$$\mathbf{Reg}(\hat{\pi}) := v(\pi^*) - v(\hat{\pi}). \tag{1}$$

The following variant of the performance difference lemma will be used throughout the paper (Duan et al., 2021). It shows that it is sufficient to control the squared Bellman error to bound regret.

**Lemma 1.** *For $f_1, ..., f_H$, let $\pi := (\pi_{f_h})_{h \in [H]}$. Then, $\mathbf{Reg}(\pi) \leq 2\sqrt{\mathcal{C}(\mu) \sum_{h \in [H]} \|f_h - T_h^* f_{h+1}\|_{\mu_h}^2}$.*

# 3 Model Selection Objectives

In this section, we state our primary model selection objectives and discuss their significance as well as challenges associated with solving them.

## 3.1 The Model Selection Problem

For a finite function class $\mathcal{F}$ that we consider here, the gold-standard regret guarantee for offline algorithms with value function approximation is

$$\mathbf{Reg}(\hat{\pi}) = \tilde{\mathcal{O}}\left(\sqrt{\mathcal{C}(\mu)\mathrm{APPROX}(\mathcal{F})} + \sqrt{\frac{\mathcal{C}(\mu)\log|\mathcal{F}|}{n}}\right), \tag{2}$$

where $\mathrm{APPROX}(\mathcal{F}) := \max_{h \in [H], f' \in \mathcal{F}} \min_{f \in \mathcal{F}} \|f - T_h^* f'\|_\mu^2$ is the completeness error of the class $\mathcal{F}$ (Chen and Jiang, 2019). This is achieved, for example, by the Fitted Q-Iteration (FQI) algorithm. If we were using infinite classes, we would replace $\log|\mathcal{F}|$ with a suitable notion of complexity such as pseudodimension. Such bounds naturally exhibit a trade-off: larger classes may have a better chance of keeping $\mathrm{APPROX}(\mathcal{F})$ close to zero[7] but require more data to minimize the estimation error. Small classes face the opposite problem.

**Definition 1.** *A class $\mathcal{F}$ is complete if $\mathrm{APPROX}(\mathcal{F}) := \max_{h \in [H], f' \in \mathcal{F}} \min_{f \in \mathcal{F}} \|f - T_h^* f'\|_\mu^2 = 0$.*

The objective of model selection is to achieve refined regret bounds that balance approximation and estimation error. To this end, we assume that the learner is presented with not just a single model class $\mathcal{F}$, but rather a nested sequence of $M$ classes $\mathcal{F}_1 \subseteq ... \subseteq \mathcal{F}_M$. Nested model classes are common in both supervised learning and offline RL. For example, one often starts with an extremely large class $\mathcal{F}$ and then considers restrictions of $\mathcal{F}$ to an increasing sequence $\mathcal{F}_1 \subseteq ... \subseteq \mathcal{F}_M = \mathcal{F}$. In a linear setting, this could correspond to trying to find a subset of candidate features that are sufficient to solve the problem.

Since the approximation error is typically unknown *a priori*, we aim to design an algorithm capable of selecting a good class in a data-dependent manner. In particular, we would like to achieve *oracle inequalities* reflecting that we can compete with the performance of an oracle that has this knowledge in advance.

Our primary objective is to compete with the *minimally complete* model class.

**Problem 1.** *Let $k_* = \min\{k \in [M] : \mathcal{F}_k \text{ is complete}\}$. Find $\hat{\pi}$ with $\mathbf{Reg}(\hat{\pi}) = \tilde{\mathcal{O}}(\sqrt{\mathcal{C}(\mu)\log(|\mathcal{F}_{k_*}|)/n})$.*

Here, $\mathcal{F}_{k_*}$ is the smallest class that satisfies completeness on the data distribution. Such oracle inequalities are common in model selection for online bandits and RL (Foster et al., 2019) – albeit they are generally not rate-optimal in that literature. In particular, Problem 1 states the regret bound should achieve the same dependence on $\log|\mathcal{F}_{k_*}|$ and $n$, as would an optimal offline algorithm using a single class with $k = k_*$. In other words, we do not tolerate any worse dependence on either quantity such as $\tilde{\mathcal{O}}(1/n^{1/4})$ rates and other lower order terms.

We are also interested in a *robustness* when $k_*$ may not exist, i.e. all $\mathcal{F}_k$ have some approximation error.

---

[7]In contrast to realizability, this intuition of monotonicity of $\mathrm{APPROX}(\mathcal{F})$ is not universally true for completeness. Adding functions to the class $\mathcal{F}$ might actually *increase* $\mathrm{APPROX}(\mathcal{F})$. However, it remains a useful heuristic. In Appendix A, we discuss how model selection in this setting is not possible without nestedness.

**Problem 2.** *Define the global completeness error as $\xi_k := \max_{h\in[H], f'\in\mathcal{F}_M} \min_{f\in\mathcal{F}_k} \|f - T_h^* f'\|_{\mu_h}^2$. Find $\hat{\pi}$ so that $\mathbf{Reg}(\hat{\pi}) = \tilde{\mathcal{O}}\left(\min_{k\in[M]}\left\{\sqrt{\mathcal{C}(\mu)\xi_k} + \sqrt{\mathcal{C}(\mu)\log(|\mathcal{F}_k|)/n}\right\}\right)$*

Note that $\xi_k \geq \textsc{Approx}(\mathcal{F}_k)$ by definition. For the estimation error, however, the guarantee remains rate-optimal. We remark that a solution to one of the above problems does not immediately imply a solution to the other. For example, a class $\mathcal{F}_k$ may be complete, but $\xi_k$ can still be large. Perhaps surprisingly, our proposed algorithm will be able to handle both problems *simultaneously* without knowledge of whether $k_*$ exists, thus achieving the $\min$ of both oracle inequalities.

### 3.2 Limitations of Prior Approaches

We now review some of the core challenges involved in solving the above problems. There are a number of seemingly natural approaches to model selection in RL that are surprisingly unable to produce satisfactory results, at least off-the-shelf.

**Adaptive offline policy evaluation**   The most natural approach, to which we have alluded in the introduction, is to first compute $\hat{\pi}_k$ with a base algorithm using function class $\mathcal{F}_k$, for each $k\in[M]$. Then, one can estimate $v(\hat{\pi}_k)$ using an off-the-shelf offline policy evaluation approach such as fitted $Q$-evaluation (Munos and Szepesvári, 2008; Duan et al., 2020), DICE methods (Nachum et al., 2019; Dai et al., 2020; Zhan et al., 2022), marginalized importance estimators (Xie et al., 2019), or doubly robust estimators (Jiang and Li, 2016; Thomas and Brunskill, 2016). Then one simply picks the $\hat{\pi}_k$ with the best estimated value. The main drawback of this approach is that nearly all of the above methods require selecting a model class to perform the estimation,[8] and it is unclear how to balance the estimation and approximation error optimally to compete with the oracle. One possible solution is to employ the adaptive estimator of Su et al. (2020), which takes as inputs a sequence of offline estimators and known upper bounds on their deviations and returns an estimator that competes with the best one. This is precisely the approach taken by Lee et al. (2021b) for linear contextual bandits. However, for general function classes in RL, there is no obvious way to compute the analogous deviation bounds, which oftentimes depend on the unknown quantity $\mathcal{C}(\mu)$. Since these bounds are required by the adaptive estimator as inputs, we are yet again left with unknown hyperparameters to tune.

**Bellman error estimators**   Recall we are focusing on base offline RL algorithms that attempt to minimize the squared Bellman error of objective. Therefore, one might ask whether it is possible to estimate the Bellman errors (e.g. with the validation dataset) and compare the model classes using the Bellman error as a proxy. Consider, for example, FQI which iteratively minimizes the squared Bellman error:

$$\hat{f}_h = \underset{f\in\mathcal{F}_k}{\arg\min}\hat{\mathbb{E}}_{D_h}\left[\left(f(x,a) - r - \max_{a'}\hat{f}_{h+1}(x',a')\right)^2\right],$$

where we use $\hat{\mathbb{E}}_{D_h}$ to denote the empirical mean calculated with samples from the dataset $D_h$. Presumably, we could simply choose the model class $\mathcal{F}_k$ that has the smallest cumulative squared error. The main issue with this approach is the classic double-sampling problem (Baird, 1995; Duan et al., 2021): the standard estimator of the Bellman error is biased because of the empirical version of the Bellman operator $T^*$. With this selection criterion, we will end up favoring model classes that also induce low variance of the *regression targets*, given by $r + \hat{f}_{h+1}(x')$ at step $h$ because the expectation is

$$\mathbb{E}_{\mu_h}\left[\left(\hat{f}_h(x,a) - r - \hat{f}_{h+1}(x')\right)^2\right] = \|\hat{f}_h - T^*\hat{f}_{h+1}\|_{\mu_h}^2 + \mathbb{E}_{\mu_h}\left[\underset{x'\sim\mathbb{P}(\cdot|x,a)}{\text{var}}\left(\hat{f}_{h+1}(x')\right)\right].$$

We want to choose a class $\mathcal{F}_k$ to minimize only the first term on the right-hand side, summed over $h\in[H]$, following Lemma 1. However, the second term is generally unknown. One could assume there is a sufficiently powerful class $\mathcal{G}$ such that $T^*f\in\mathcal{G}$ for all $f\in\mathcal{F}$ (Chang et al., 2022). But there remains a question of how to select the class $\mathcal{G}$ to trade off approximation error and estimation error, creating another unsolved model selection problem. In the same vein, another approach we might consider is the BVFT algorithm of Xie and Jiang (2021) to select among the $f^k$ learned by the base algorithm. This solves the model selection problem but BVFT has a slow $\mathcal{O}(1/n^{1/4})$ dependence. It also, in theory, requires that a discretization parameter is set based on a concentrability coefficient stronger than $\mathcal{C}(\mu)$,

---

[8]For marginalized importance sampling, the guarantee is not strong enough to compete with the oracle.

which is typically unknown. Perhaps most conceptually related is past work which compares Bellman errors of finer-grained state abstraction functions on the Q-function computed on coarser-grain state abstraction (Jiang et al., 2015). This work provided regret bounds on in the discrete state and action setting where the models are varying levels of state abstractions. However, this critically depends on the discrete state and action setting. Our work shows how a similar idea can be used in the value function approximation setting, with substantially different tools and analysis techniques.

**Representation Learning** Model selection resembles objectives in representation learning for RL (Agarwal et al., 2020; Modi et al., 2021). However, we cannot simply adapt such algorithms since they are either insensitive to the class complexities or they require stronger realizability assumptions. Understanding the connections of these problems would be interesting for future work.

## 4   MODBE Algorithm

Having introduced the model selection objectives, we now present our main result, a novel model selection algorithm for offline RL that provably achieves the aforementioned oracle inequalities. We first give an intuitive sketch of the approach. As a thought experiment, we will consider the case when $M = 2$ and a minimally complete class $\mathcal{F}_{k_*}$ exists.[9] We will also ignore logarithmic factors and $H$ dependence for now. A key algorithmic idea is that we will first start optimistically by guessing that $k_* = 1$. Running a base algorithm like FQI with $\mathcal{F}_1$ on training data returns the functions $f_1, ..., f_H$, which, with high probability, satisfy $\sum_h \|f_h - T_h^* f_{h+1}\|_\mu^2 = \tilde{\mathcal{O}}\left(\frac{\mathcal{C}(\mu)\log(|\mathcal{F}_1|)}{n}\right)$ if $k_*$ equals 1. Given these functions, we can pose a square loss regression problem where the regression targets (i.e., the "y's" of the regression problem) are given by the empirical Bellman updates:

$$\hat{L}_h(g, f_{h+1}) = \tfrac{1}{n}\sum_{(x,a,r,x')\in D_h}(g(x,a) - r - f_{h+1}(x'))^2.$$

Let $L_h(f, g) := \mathbb{E}_{\mu_h}\left[\hat{L}_h(f, g)\right]$. Solving this regression problem for each $h$ over the class $\mathcal{F}_2$ will generate $g_1, ..., g_H \subseteq \mathcal{F}_2$. The key insight is that the sequences $(f_h)_h$ and $(g_h)_h$ are both trying to minimize the same empirical square loss function with the same regression targets: $r + f_{h+1}(x')$. Unlike the Bellman error estimators from the previous section that incur biases, the losses $L_h(f_h, f_{h+1})$ and $L_h(g_h, f_{h+1})$ are comparable and estimable from a validation set. By nestedness of $\mathcal{F}_1 \subseteq \mathcal{F}_2$, $\mathcal{F}_2$ cannot have more approximation error on this regression problem. Provided we can get a good estimate of generalization errors $L_h(f_h, f_{h+1})$ and $L_h(g_h, f_{h+1})$ with validation data, this naturally brings forth the following *generalization test*: if

$$L_h(g_h, f_{h+1}) < L_h(f_h, f_{h+1}) - \tilde{\mathcal{O}}\left(\frac{\log(|\mathcal{F}_1|)}{n}\right) \tag{3}$$

reject $\mathcal{F}_1$ and pick $\mathcal{F}_2$. Otherwise pick $\mathcal{F}_1$. That is, a switch will occur not when $\mathcal{F}_2$ performs only marginally better than $\mathcal{F}_1$, but when it performs *substantially* better as measured by the generalization error that we see for both $f_h$ and $g_h$ on this regression problem. If (3) holds, then there is reason to believe that $\mathcal{F}_1$ is not complete, making $\mathcal{F}_2$ the right choice. Crucially, the test only checks for generalization error, so the tolerance term on the right side goes as $\tilde{\mathcal{O}}(\log(|\mathcal{F}_1|)/n)$, which is the correct rate for this problem. Thus, if the test turns out to be wrong, we will only lose additive factors of the correct rate.

### 4.1   Full Algorithm

The full algorithm, MODBE (Model Selection via Bellman Error), is presented in Algorithm 1. While the underlying principle described just above is similar, MODBE must handle a number extensions that complicate the algorithm such as dealing with general $M$, accounting for proper estimation errors, and being robust to the case when $k_*$ does not exist. Interestingly, the fundamental algorithmic idea remains the same – only the tolerances change and it loops over the model classes.

MODBE takes as input a base offline RL algorithm (such as FQI), the model classes $\mathcal{F}_1 \subseteq ... \subseteq \mathcal{F}_M$, and the offline dataset $D \in [H]$. The dataset is split randomly into a training set $D_{\text{train}}$ and a validation set $D_{\text{valid}}$. The algorithm begins optimistically, starting with the candidate model class $k = 1$ and running the base algorithm with $\mathcal{F}_k$ on the training dataset to generate the candidate functions $f$. We

---

[9]While the full algorithm requires minimal changes beyond this intuition there are challenges in the proof. For general $M$, we cannot guarantee the class returned will be the correct – but it may have controllable approximation error. When $k_*$ does not exist, there is a chance to "skip" the best model class.

---

**Algorithm 1** Model Selection via Bellman Error (MODBE)

---

1: **Input**: Offline dataset $D = (D_h)$ of $n$ samples for each $h \in [H]$, Base algorithm $\mathcal{B}$, function classes $\mathcal{F}_1 \subseteq ... \subseteq \mathcal{F}_M$, failure probability $\delta \leq 1/e$, and estimation error function $\omega$ for $\mathcal{B}$.
2: Let $n_{\text{train}} = \lceil 0.8 \cdot n \rceil$ and $n_{\text{valid}} = \lfloor 0.2 \cdot n \rfloor$ and split the dataset $D$ randomly into $D_{\text{train}} = (D_{\text{train},h})$ of $n_{\text{train}}$ samples and $D_{\text{valid}} = (D_{\text{valid},h})$ of $n_{\text{valid}}$ samples for each $h \in [H]$.
3: Set $\zeta := \frac{96H^2\log(16M^2H/\delta)}{n_{\text{valid}}}$
4: Initialize $k \leftarrow 1$.
5: **while** $k < M$ **do**
6:     $(f_h)_{h \in [H]} \leftarrow \mathcal{B}(D_{\text{train}}, \mathcal{F}_k, \delta/4M)$
7:     **for** $k' \leftarrow k+1,...,M$ **do**
8:         Set $\alpha := \max\left\{\omega_{n_{\text{train}}, \delta/4M}(\mathcal{F}_{k'}), \frac{200H^2\log(8M^2H|\mathcal{F}_{k'}|/\delta)}{n_{\text{train}}}\right\}$
9:         Set $\text{TOL} := 2\alpha + 2\zeta + \omega_{n_{\text{train}}, \delta/4M}(\mathcal{F}_k)$
10:         Minimize squared loss on training set for all $h \in [H]$ with regression targets from class $k$:

$$g_h \leftarrow \underset{g \in \mathcal{F}_{k'}}{\arg\min} \quad \hat{L}_h(g, f_{h+1}) := \frac{1}{n_{\text{train}}} \sum_{(x,a,r,x') \in D_{\text{train},h}} (g(x,a) - r - f_{h+1}(x'))^2 \quad (4)$$

11:         Compute squared loss using the validation set for all $h \in [H]$ as a function of $f$:

$$\tilde{L}_h(f, f_{h+1}) = \frac{1}{n_{\text{valid}}} \sum_{(x,a,r,x') \in D_{\text{valid},h}} (f(x_h, a_h) - r_h - f_{h+1}(x'))^2 \quad (5)$$

12:         **if** $\tilde{L}_h(g_h, f_{h+1}) < \tilde{L}_h(f_h, f_{h+1}) - \text{TOL}$ for any $h \in [H]$ **then**
13:             $k \leftarrow k+1$
14:             goto Line 5.
15:         **end if**
16:     **end for**
17:     goto Line 19
18: **end while**
19: **return** $\hat{\pi} = (\pi_{f_h})_{h \in [H]}$

---

retrain on the empirical square loss using a class $k' > k$ by regressing to target values $r + f_{h+1}(x')$. This amounts to solving a sequence of $H$ least squares regression problems using class $k'$, yielding the functions $g_h$. Since $f_h$ and $g_h$ are attempting to solve the *same* regression problem (with the same target values), we can compare their performance on this shared squared loss objective $\tilde{L}$ with validation data. We use a *generalization error test* in Line 12 to decide whether to keep using class $k$. If the test fails and it is discovered that the larger model class $\mathcal{F}_{k'}$ is able to achieve substantially smaller loss than $\mathcal{F}_k$, then we move to a larger model class $k \leftarrow k+1$. The process is repeated until all classes are exhausted or no model class $k'$ offers a big enough improvement over $k$ to cause the test to fail.

## 4.2 Rate-Optimal Oracle Inequalities

We show that MODBE is able to achieve both Problems 1 and 2 simultaneously. We start with a generic version of the theorem stated in terms of an assumed performance bound $\omega$ on the base algorithm. We will presently instantiate the base algorithm with FQI, showing that it precisely achieves the oracle inequalities with the correct rates.

**Definition 2.** *Let $\mathcal{B}$ be a base offline RL algorithm for value function approximation that takes as input a model class $\mathcal{F}$, an offline dataset $D$ of $n$ samples for each $h \in [H]$, and a failure probability $\delta$. For $\beta > 0$ and a function $\omega$, we say that $\mathcal{B}$ is $(\beta, \omega)$-regular if (1) $\omega$ is a known real-valued function of $n \in \mathbb{N}$, $\delta \in \mathbb{R}$, and $\mathcal{F}_k$, and it satisfies $\omega_{n,\delta}(\mathcal{F}_k) \leq \omega_{n,\delta}(\mathcal{F}_{k'})$ for all $k' \geq k$; (2) $\mathcal{B}(D, \mathcal{F}_k, \delta)$ returns $(f_h)_{h \in [H]} \subseteq \mathcal{F}_k$ such that $f_{h+1}$ is independent of $D_h$ and*

$$P\left(\max_{h \in [H]} \|f_h - T_h^* f_{h+1}\|_{\mu_h}^2 \leq \beta \cdot \text{APPROX}(\mathcal{F}_k) + \omega_{n,\delta}(\mathcal{F}_k)\right) \geq 1 - \delta. \quad (6)$$

Here, $\beta$ scales the approximation error and $\omega$ represents the estimation error. Generally, we will have $\omega_{n,\delta}(\mathcal{F}) = \tilde{\mathcal{O}}(\log(|\mathcal{F}|/\delta)/n)$ (see Lemma 2 for FQI). We hope to achieve a bound that matches what the base algorithm would achieve had $k_*$ been known in advance, up to additive terms of $\sqrt{\log(|\mathcal{F}_{k_*}|)/n}$.

Our primary theorem addresses Problem 1 using an arbitrary base algorithm.

**Theorem 2.** *Let $\mathcal{B}$ be an $(\beta, \omega)$-regular algorithm and suppose that $k_*$ (defined in Problem 1) exists. Let $\iota = \log(M^2 H/\delta)$. Then, for some absolute constant $C > 0$, Algorithm 1 with inputs $D$, $\mathcal{B}$, $\mathcal{F}_1 \subseteq ... \subseteq \mathcal{F}_M$, $\omega$, and $\delta \leq 1/e$ outputs $\hat{\pi}$ such that, with probability at least $1 - \delta$,*

$$\textbf{Reg}(\hat{\pi}) \leq C \cdot \sqrt{\mathcal{C}(\mu) H \left( \omega_{n_{train}, \delta/4M}(\mathcal{F}_{k_*}) + \frac{H^2 (\log |\mathcal{F}_{k_*}| + \iota)}{n} \right)} \tag{7}$$

The above theorem shows a regret bound scaling with the square root of the error term $\omega$ of the base algorithm $\mathcal{B}$ plus a $\tilde{\mathcal{O}}(\log(|\mathcal{F}_{k_*}|)/n)$ estimation error. Importantly, as stated in Problem 1, the statistical complexity depends only on $\mathcal{F}_{k_*}$ and not any of the larger classes. For concreteness, we now instantiate Theorem 2 with a standard finite-horizon FQI (Duan et al., 2020) base algorithm, which satisfies Definition 2 with $\omega_n(\mathcal{F}) = \hat{\mathcal{O}}(\log|\mathcal{F}|/n)$. This in turn translates to the desired rate-optimal oracle inequalities.

**Lemma 2.** *Consider the FQI algorithm (stated in Appendix C for completeness). For a model class $\mathcal{F}$, FQI is a $(3, \omega)$-regular base algorithm with $\omega_{n,\delta}(\mathcal{F}) = \mathcal{O}\left( \frac{H^2 \log(H|\mathcal{F}|/\delta)}{n} \right)$.*

By plugging this classic result in Theorem 2 as the base algorithm, we arrive at a solution to Problem 1.

**Corollary 1.** *Let $\mathcal{B}$ be instantiated with FQI (Algorithm 3 in Appendix C). Define $\iota = \log(M^2 H/\delta)$. Then, under the same conditions as Theorem 2, there is an absolute constant $C > 0$ such that, with probability at least $1 - \delta$, Algorithm 1 outputs $\hat{\pi}$ satisfying*

$$\textbf{Reg}(\hat{\pi}) \leq C \cdot \sqrt{\frac{\mathcal{C}(\mu) H^3 (\log|\mathcal{F}_{k_*}| + \iota)}{n}}. \tag{8}$$

The proof of Theorem 2 (Corollary 1) follows the intuition from the start of this section. The proof shows (1) MODBE will never return a value of $k$ that exceeds $k_*$ and (2) if MODBE returns $k < k_*$, then the approximation error must be small. A key novelty is recognizing that the generalization test in Line 12, which compares the errors classes on the same regression problem, can be used to prove both (1) and (2).

**Robustness** We show that the same Algorithm 1 simultaneously achieves the desired robustness result of Problem 2 when $k_*$ does not exist without any modification.

**Theorem 3.** *Under the same conditions as Theorem 2, if $k_*$ does not exist, there exists an absolute constant $C > 0$ such that, with probability at least $1 - \delta$, Algorithm 1 outputs $\hat{\pi}$ satisfying*

$$\textbf{Reg}(\hat{\pi}) \leq C \cdot \min_{k \in [M]} \left\{ \sqrt{\mathcal{C}(\mu) H \left( \beta \cdot \xi_k + \omega_{n_{train}, \delta/4M}(\mathcal{F}_k) + \frac{H^2 (\log|\mathcal{F}_k| + \iota)}{n} \right)} \right\}. \tag{9}$$

We can use Lemma 2 to see a solution to Problem 2 with an instantiation of FQI.

**Corollary 2.** *Under the same conditions as Corollary 1, there is an absolute constant $C > 0$ such that, with probability at least $1 - \delta$, Algorithm 1 outputs $\hat{\pi}$ satisfying*

$$\textbf{Reg}(\hat{\pi}) \leq C \cdot \min_{k \in [M]} \left\{ \sqrt{\mathcal{C}(\mu) H \xi_k} + \sqrt{\frac{\mathcal{C}(\mu) H^3 (\log|\mathcal{F}_k| + \iota)}{n}} \right\} \tag{10}$$

The guarantees that solve Problems 1 and 2 are achieved simultaneously, meaning that we do not require knowledge of whether $k_*$ exists and we can automatically get the best of both. The proof of Theorem 3 (Corollary 2) is more involved, but still crucially leverages the generalization test in Line 12. Here, we allow $k$ to exceed the minimal index sometimes. We then use the fact that class $(k-1)$ must have failed the generalization test to argue that the estimation error of the larger class can be bounded by the unknown $\xi_{k-1}$, which is small since $k$ exceeds the index of the minimal class.

**Computational Complexity** MODBE is computationally efficient given a squared loss regression oracle. Within inner and outer loops over the model classes, a squared loss minimizer is computed on the training dataset and then functions are evaluated on the validation set. MODBE requires only $\mathcal{O}(H k_* M)$ calls to the computational oracle when $k_*$ exists (a consequence of Theorem 2) or $\mathcal{O}(H M^2)$ in the worst case. Note that algorithms for optimizing squared loss regression problems are ubiquitous in machine learning (Simchi-Levi and Xu, 2021).

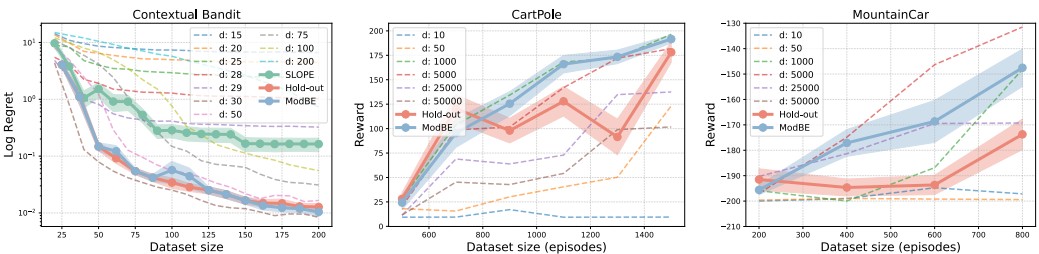

Figure 1: MODBE is evaluated on several simulated domains: a contextual bandit (left), CartPole (middle), and MountainCar (right). In CB, MODBE and Hold-out outperform SLOPE and match performance of the best model class in regret. In CartPole, both match the performance of the best model class. In MountainCar, both struggle to match the best model class, but MODBE maintains superior performance. In CB, error bands are standard error over 10 random trials. In RL, error bands are standard error over 20 random trials.

## 5 Empirical Results

The previous sections outlined the strong theoretical properties of MODBE. In this section, we ask: what practical insights can be gleaned from MODBE and its theoretical guarantees? We would like to understand if the core selection method of MODBE can be applied out-of-the-box on existing offline RL algorithms with minimal effort. We evaluated MODBE in three simulated environments with discrete actions: (1) synthetic contextual bandits (CB), (2) Gym CartPole, (3) Gym MountainCar. See Appendix D for specific details about the setups. All training and validation sets were split 80/20.

**Contextual Bandit** As a basic validation experiment, we started with the CB setting of Lee et al. (2021b) which considers a nested sequence of linear model classes with increasing dimension $d$. Without any tuning, we simply set the tolerance of MODBE to $\text{TOL}(\mathcal{F}_k, \mathcal{F}_{k'}) = \frac{d_{k'}}{n}$. Figure 1 shows the results in terms of the $\log$-regret as a function of the dataset size. We observe that both MODBE and Hold-Out (choosing the model class with the smallest error) are able to easily match the performance of the best model class while SLOPE (Lee et al., 2021b) ends up being fooled by nearby classes.

**RL Discrete Control** Our setup for the RL problems in Gym (Brockman et al., 2016) builds on top of the open-source d3rlpy framework (Seno and Imai, 2021). We used DQN (Mnih et al., 2015), which is closest to FQI. We considered model classes that were two-layer neural networks with ReLU activations and $d$ nodes in the hidden layer and varied the parameter $d$. Again, we simply set the tolerance of MODBE to $d_k/n$ motivated by pseudodimension bounds (Bartlett et al., 2019). For simplicity, we modified MODBE to work in the discounted infinite horizon setting, which can trivially be done (see Appendix D). We compared MODBE to Hold-Out, which is a seemingly sensible baseline that chooses the model class with lowest estimated Bellman error on a validation set. For deterministic settings only, this is theoretically justified. Figure 1 shows the reward as a function of the dataset size (in episodes). On CartPole, MODBE marginally outperforms Hold-Out. On MountainCar, we find that Hold-Out does surprisingly poorly while MODBE is successfully able to reject the poor model classes. We conjecture that the empirical failure of Hold-Out is possibly due to sensitivity to optimization error making the Bellman error misleading. In contrast, the generalization test of MODBE seems to be more robust.

## 6 Discussion

In this paper, we introduced a new algorithm, MODBE, for model selection in offline RL: to our knowledge it is the first to achieve rate-optimal oracle inequalities in $n$ and $\text{COMP}(\mathcal{F}_{k_*})$. A number of interesting open questions remain. (1) Are there rate-optimal procedures that can be used to select hyperparameters beyond model complexity such as learning rates, batch sizes, *et cetera*? (2) Can the ideas of MODBE be extended to more general algorithms that do not rely on Bellman error minimization? (3) For the robustness guarantee, the global completeness $\xi$ is potentially much worse than $\text{APPROX}(\mathcal{F})$. Is it possible to achieve a robust oracle inequality of the form $\mathcal{O}(\min_k \sqrt{\text{APPROX}(\mathcal{F}_k) + \log|\mathcal{F}_k|/n})$ when $k_*$ does not exist? We believe these questions are of great practical and theoretical importance for understanding how to effectively evaluate and select models in offline RL.

## Acknowledgments and Disclosure of Funding

We thank Annie Xie and Yannis Flet-Berliac for help and advice with experiments and anonymous reviewers for their valuable feedback. JNL is supported by the NSF GRFP. This work was also supported in part by NSF Grant #2112926.

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
