# OpenReview forum: "Oracle Inequalities for Model Selection in Offline Reinforcement Learning"
_NeurIPS.cc/2022/Conference — NeurIPS 2022 Accept_

### Official Review · Reviewer_cD9U · 2022-06-26

**Rating:** 7
**Confidence:** 4
**Soundness:** 4 excellent
**Presentation:** 3 good
**Contribution:** 3 good

**Summary:**

This paper considers the model selection problem for offline RL where the learner is given a nested set of model classes and must choose which one to use completely offline without interacting with the environment. The paper extends previous work from the contextual bandit setting to the RL setting and achieves minimax optimal rates up to log factors for a novel algorithm ModBE. Importantly, ModBE relies on a one-sided generalization test where regressors from two models classes are compared on the same regression problem induced by the Bellman targets of the smaller model class in the nested ordering. Some empirical results are presented on small domains.

**Questions:**

Is it clear that we need to make assumption 1 or just that a simple assumption with one comparator policy is not sufficient? If we need assumption 1, how important are optimal rates if dependence on H is exponential regardless? Should we consider a more structured set of problems to avoid these issues and yield more practical insights?

Can you provide empirical evidence where ModBE is clearly beneficial? Or even a toy problem where hold-out fails but ModBE succeeds?

**Limitations:**

The paper is sufficiently abstract that I think it is ok that it does not explicitly address ethical concerns in the main text.

**Strengths And Weaknesses:**

Strengths:

1. The technique of applying the one-sided test on pairs of model classes for the same Bellman backup regression problem is clever and novel. The novel technique could be useful to the community as it is possible that the core idea could find uses in other problems around RL.

2. The proofs seem to be correct, although I haven't checked everything in the appendix carefully.

Weaknesses:

1. In general it seems that one should expect C(\mu) to be exponential in H. The lower bound from LTND21 says that dependence on a comparator policy is not possible, so the authors argue that this term is fundamental. But at that point, the main results of the paper seem to really only apply to short horizon problems. For any longish horizon, this constant will likely be enormous and the optimal rates may not really matter. It is still nice to have the optimal rates in n, but this seems like a serious issue that is sort of glossed over in the paper. It is possible that the problem as stated really is just too hard and that we should instead be considering more structured settings where the exponential lower bound could be avoided rather than proving bounds that are exponential.

2. The experiments do not seem to show much benefit from the ModBE algorithm over the naive hold-out baseline. On the bandit and cartpole problems there is no noticeable difference at all. On mountain car ModBE seems slightly better, but never enough that the errorbars do not overlap with the baseline (and it seems that performance in the dataset is not good enough to learn a decent policy, no matter the model class). In my mind, these experiments cannot be seen as evidence of the usefulness of ModBE over the much simpler (both conceptually and implentation-wise) baseline.

3. In practice, the ModBE algorithm itself has a hyperparameter in the form of the tolerance value. In the theory this can be set to some well-motivated value, but in practice it is often the case that those values will be overly conservative (as in most RL theory to practice conversions). For example, in the experiments the authors use a heuristic to set the tolerance based on pseudodimension bounds, but this may not be accessible with more realistic (i.e. larger neural net) model classes and then the parameter would need to be tuned. There is no discussion about sensistivity to this choice in practice and it seems like a serious issue for an algorithm that attempts to do model selection to potentially be hyperparameter sensitive itself.

Summary:

Overall, I think the paper studies an important problem and presents a clever algorithm and proof, but am wary of the significance of the contribution on a few levels: it seems that the assumptions effectively render the bound exponential in most realistic problems and from a practical perspective the paper does not provide clear evidence on the usefulness or robustness of the method.

---

> ### Author Response · Authors · 2022-08-02
> **Response**
>
> Thank you for your review and feedback to improve the paper. Please find responses below.
>
> > In general it seems that one should expect C(\mu) to be exponential in H.
>
>
> While $C(\mu)$ is exponential in $H$ in the worst case (such as combination lock problems), there are many natural settings where $C(\mu)$ has no horizon dependence such as many mixing MDPs. Assuming any of the natural conditions that imply that $C(\mu)$ is small would be a _stronger_ assumption, so would immediately be compatible with our results. We will add a discussion of this to the paper.
>
> Moreover, the dependence on $C(\mu)$ in our work is the default assumption in offline RL theory, since non-trivial results are essentially information-theoretically impossible without further assumptions [1].
>
>
> [1] Chen, Jinglin, and Nan Jiang. "Information-theoretic considerations in batch reinforcement learning." International Conference on Machine Learning. PMLR, 2019.
>
> > Is it clear that we need to make assumption 1 or just that a simple assumption with one comparator policy is not sufficient?
>
> For single model class offline RL, some recent papers have tried to soften this assumption on $C(\mu)$ by assuming good coverage of the dataset only on a comparator’s distribution [2, 3]. But coverage is still being assumed and it does not preclude exponential in H dependence in the worst case. Furthermore, [4] has shown that, for model selection, such results are not information-theoretically possible.
>
> [2] Jin, Ying, Zhuoran Yang, and Zhaoran Wang. "Is pessimism provably efficient for offline rl?." International Conference on Machine Learning. PMLR, 2021.
>
> [3] Xie, Tengyang, et al. "Bellman-consistent pessimism for offline reinforcement learning." Advances in neural information processing systems 34 (2021): 6683-6694.
>
> [4] Lee, Jonathan, et al. "Model selection in batch policy optimization." International Conference on Machine Learning. PMLR, 2022.
>
> > If we need assumption 1, how important are optimal rates if dependence on H is exponential regardless? Should we consider a more structured set of problems to avoid these issues and yield more practical insights?
>
> Please note that our bounds are not inherently exponential. The assumption that $C(\mu)$ is small _is_ a structural assumption. Many natural conditions imply that $C(\mu)$ is small, so to assume those instead would be making a stronger assumption and would thus narrow the applicability of the results. The technical focus of this paper is the model selection problem in a general and well-established theoretical setting, but we agree that better understanding practically relevant structured assumptions is an important area of study in general.
>
> > Can you provide empirical evidence where ModBE is clearly beneficial? Or even a toy problem where hold-out fails but ModBE succeeds?
>
> Thanks for the suggestion. In running the algorithms for more trials, we see that the difference becomes much more stark. Please see the revised paper in Line 321. No changes were made besides changing the number of trials from 5 to 20 for each one to reduce noise for the RL experiments. The CB experiments are the same.
>
> The contribution of this paper is primarily theoretical and the purpose of the experiments is to understand whether the theoretical intuition carries forward in practice. Note that hold-out does not satisfy any theoretical guarantees, except in special cases.

---

> > ### Comment · Reviewer_cD9U · 2022-08-07
> > **Response to authors**
> >
> > Thanks for the detailed response. I think I was being unfair in my criticism about main assumption and the rebuttal does a good job explaining this. The improved experimental results are also useful to see a significant benefit of the proposed algorithm in at least one of the tasks.
> >
> > I will increase my score to accept to reflect my better appreciation of the theoretical results and improved experimental results.

---

> > > ### Author Response · Authors · 2022-08-09
> > > **Thank you**
> > >
> > > Dear Reviewer,
> > >
> > > Thank you for your reply! We are happy to discuss more in the time remaining if you have additional questions.

---

### Official Review · Reviewer_ewUJ · 2022-07-11

**Rating:** 3
**Confidence:** 3
**Soundness:** 2 fair
**Presentation:** 2 fair
**Contribution:** 1 poor

**Summary:**

This paper proposes an algorithm for offline reinforcement learning.

**Questions:**

What is the relation and difference of this paper and other works for offline RL, including [1], [2], and [3].

[1]. Xie, Tengyang, et al. "Bellman-consistent pessimism for offline reinforcement learning." Advances in neural information processing systems 34 (2021): 6683-6694.

[2]. Zhan, Wenhao, et al. "Offline reinforcement learning with realizability and single-policy concentrability." Conference on Learning Theory. PMLR, 2022.

[3]. Uehara, Masatoshi, and Wen Sun. "Pessimistic model-based offline reinforcement learning under partial coverage." arXiv preprint arXiv:2107.06226 (2021).


**Limitations:**

The relation and difference of their work and other works in offline RL is unmentioned.

**Strengths And Weaknesses:**

Strength: This paper provide an algorithm and upper bound its suboptimality.

Weakness. The relation and difference of their work and other works in offline RL is unmentioned.

---

> ### Author Response · Authors · 2022-08-02
> **Response**
>
> Thank you for pointing out additional references. We will add them to the paper. However, while related to offline reinforcement learning, these papers are tangential to the purpose of the present work. In this work, we are studying the model selection problem, whereas the referenced papers study policy optimization. [1] and [3] study the problem of leveraging pessimism to soften coverage assumptions. In Sec 2.1, we discuss that analogous results are not possible for model selection. [2] studies the problem of achieving provable guarantees without completeness, under the two different realizability assumptions. This was mentioned in the paper. We will revise the paper to make the distinction between model selection and policy optimization clearer.

---

> ### Author Response · Authors · 2022-08-09
> **Follow up to response**
>
> Dear Reviewer,
>
> Has our response addressed your concerns? We are happy to discuss more in the time remaining.

---

### Official Review · Reviewer_QYVr · 2022-07-13

**Rating:** 4
**Confidence:** 4
**Soundness:** 3 good
**Presentation:** 3 good
**Contribution:** 3 good

**Summary:**

This paper presents an algorithm for model selection with nested model classes when used with value function approximation for offline RL. The bound achieved by the proposed algorithm is minimax rate-optimal oracle inequalities (upto logarithmic factors).

**Questions:**

Pretty much all my questions are summarized in the weaknesses above.

**Limitations:**

I'd recommend the authors consider any unintended implications of their paper's result, although, I do not see any specific limitations myself.

**Strengths And Weaknesses:**

Strengths:
- The problem and the related algorithm are new to my knowledge
- The paper is well written
- The theory is accompanied by some reasonable experimental support.

Weaknesses:
- The main issue I have with this paper is how do we obtain a set of nested function classes where such an ordering is actually satisfied? Clearly, the authors recognize this issue in footnote 3, and this, in a sense breaks much of what is developed in the paper since the paper posits the existence of nested model classes that satisfy the approximation error-statistical complexity trade-off. These issues are pretty much mirrored in existing works, where there is a need to select a model class for performing estimation and it is unclear how to tradeoff approximation and estimation error. I would recommend the authors discuss this in greater detail.
- Another issue i see is while the authors stress schemes that achieve rate-optimality, the dependence on the approximation error appear to be sub-optimal, and this is the dominating term (particularly in the limit).
- The experimental results are limited at best. I would recommend that these be expanded in scope to consider continuous control tasks (e.g. with D4RL benchmark suite) since many methods used for these tasks rely on variants of actor-critic methods. I do like the paper's result, although i am trying to understand if the authors could compensate for what appears to be non-trivial limitations in the problem setup through more extensive empirical studies.

---

> ### Author Response · Authors · 2022-08-02
> **Response**
>
> Thank you for your review and feedback to improve the paper. Please find responses below.
>
> > The main issue I have with this paper is how do we obtain a set of nested function classes where such an ordering is actually satisfied?
>
> To clarify, we do _not_ assume an ordering on Approx($\mathcal{F}_k$); we only require nested functions. We chose the nested function assumption as it is standard for model selection in supervised learning (cf. Bartlett et al 2002), and, algorithmically, it is straightforward to satisfy, for example, in linear regression by adding features to a feature vector.
>
> As you note, nested functions may not have a decreasing approximation error due to completeness. This is a challenge identified time after time in the offline RL literature (e.g., [1, 2]). While there are few solutions, all that we know of come with drawbacks of their own. In fact, a key contribution of our paper is to ease this problem: Theorem 1 shows we can compete with the smallest model class that satisfies completeness (without knowledge of which class satisfies completeness). This is a significant advantage over most existing works, which make the assumption that completeness is already satisfied (or at least the error is very small) for all model classes.
>
>
> [1] Chen, Jinglin, and Nan Jiang. "Information-theoretic considerations in batch reinforcement learning." International Conference on Machine Learning. PMLR, 2019.
>
> [2] Foster, Dylan J., et al. "Offline reinforcement learning: Fundamental barriers for value function approximation." arXiv preprint arXiv:2111.10919 (2021).
>
>
>
> > The dependence on the approximation error appear to be sub-optimal, and this is the dominating term
>
> This was unclear. The main theoretical contribution focuses on the setting where completeness is satisfied by at least one class. In such settings, we can compete with this complete class without knowing it ahead of time and thus incur no additional penalty due to approximation and maintain regret that is comparable with the oracle. This is better than any existing results, to our knowledge. The second part of the theorem can be viewed as a robustness property of the algorithm: when completeness is not satisfied by any class, non-trivial guarantees can still be obtained, but the approximation error is sub-optimal. This is, however, secondary to the main result and we will revise the paper to make this clear. Furthermore, our empirical results suggest that we can still achieve good performance even if this assumption is not completely satisfied.

---

> ### Author Response · Authors · 2022-08-09
> **Follow up to response**
>
> Dear Reviewer,
>
> Has our response addressed your concerns? We are happy to discuss more in the time remaining.

---

> ### Comment · Reviewer_QYVr · 2022-08-09
> **Thank you for your clarifications**
>
> While the theory results are certainly interesting, the authors agree with the issue that the approximation error need not decrease when having nested function classes. I view this as an issue that dilutes this paper's contributions since I am unsure how broadly this paper's contributions are useful in practice.
>
> If the authors can clarify by pointing to any empirical study in the literature on well studied benchmark offline RL datasets on the feasibility of this assumption, that can help address some of these concerns. Another way to mitigate these concerns is to have a more expansive set of experimental results by considering well studied benchmarks (e.g. the D4RL suite)  to show how these results manifest in offline RL (in practice).  I believe such an addition can greatly strengthen this paper's contributions. As it stands, however, I will retain my score and thank the authors for their detailed response and their willingness to discuss further.

---

> > ### Author Response · Authors · 2022-08-09
> > **Thanks for your response**
> >
> > Thanks for your response. The focus of this paper is primarily the theoretical aspects and the contribution is to advance our understanding of these previously unknown topics. We understand your concerns about the approximation error, but this is also an issue dealt with by many offline RL algorithms in theory. The purpose of this paper is to provide a model selection algorithm that mitigates this problem as we can compete with best model that satisfies completeness.

---

### Meta-Review · Area_Chair_VsjM · 2022-08-27

**Recommendation:** Accept
**Confidence:** Certain

**Metareview:**

This paper studies the model selection problem in the offline RL setting. The paper focuses on a theoretical study where a sequence of nested models are provided and the algorithm ought to output a class that nearly matches the optimal one (the so-called oracle inequality). It is surprising that the proposed algorithm is simple (QVI + generalization testing) but can achieve the oracle inequality. Both the meta-reviewer and some of the reviewers believe that the paper have a solid theoretical contribution and is qualified for publication in NeurIPS.

However, in addition to the issues mentioned by the reviewers, the meta-reviewer finds that the presentation could be additionally improved. For instance, in the current form of the paper, it is hard to understand the general idea by just reading the first 8 pages -- e.g., one might simply think to estimate each model separately and then choose the best one; why this simpler algorithm does not work? It would be beneficial if the authors could provide a technical overview in the main text rather than provide it in the appendix. Other minor issues including the notations. In Def. 1, the meaning of w is not provided; w was first mentioned, then then in the equations what was used is w_{n, \delta}; also, it is said w is a function -- but it is a function of what? n, delta, and F_k?

With that being said, the recommendation of this paper is an accept. The meta-reviewer encourages the authors to incorporate the reviewers' comments and further improve the presentation.

**Award:**

No

---

### Decision · Program_Chairs · 2022-09-14

Accept